# Experimental Evaluation of New Geopolymer Composite with Inclusion of Slag and Construction Waste Firebrick at Elevated Temperatures

**DOI:** 10.3390/polym15092127

**Published:** 2023-04-29

**Authors:** Ozer Sevim, Ilhami Demir, Erdinc Halis Alakara, İsmail Raci Bayer

**Affiliations:** 1Department of Civil Engineering, Kırıkkale University, Kırıkkale 71450, Turkey; 2Department of Civil Engineering, Tokat Gaziosmanpasa University, Tokat 60150, Turkey; 3Institute of Science, Kırıkkale University, Kırıkkale 71450, Turkey

**Keywords:** geopolymer, firebrick powder, elevated temperature effect, mechanical properties, different cooling regimes

## Abstract

This study investigates the effect of elevated temperatures on slag-based geopolymer composites (SGC) with the inclusion of firebrick powder (FBP). There is a limited understanding of the properties of SGC with the inclusion of FBP when exposed to elevated temperatures and the effects of cooling processes in air and water. In this regard, in the preliminary trials performed, optimum molarity, curing temperature, and curing time conditions were determined as 16 molarity, 100 °C, and 24 h, respectively, for SGCs. Then, FBP from construction and demolition waste (CDW) was substituted in different replacement ratios (10%, 20%, 30%, and 40% by slag weight) into the SGC, with optimum molarity, curing temperature, and curing time. The produced SGC samples were exposed to elevated temperature effects at 300, 600, and 800 °C and then subjected to air- and water-cooling regimes. The ultrasonic pulse velocity, flexural strength, compressive strength, and mass loss of the SGCs with the inclusion of FBP were determined. In addition, scanning electron microscopy (SEM) analyses were carried out for control (without FBP) and 20% FBP-based SGC cooled in air and water after elevated temperatures of 300 °C and 600 °C. The results show that the compressive and flexural strength of the SGC samples are higher than the control samples when the FBP replacement ratio is used of up to 30% for the samples after the elevated temperatures of 300 °C and 600 °C. The lowest compressive and flexural strength results were obtained for the control samples after a temperature of 800 °C. As a result, the elevated temperature resistance can be significantly improved if FBP is used in SGC by up to 30%.

## 1. Introduction

Due to the existence of waste in the construction sector and its adverse environmental effects, it is desirable to reuse waste and promote sustainable waste recycling. This factor has encouraged researchers to conduct studies on the evaluation of construction and demolition waste (CDW) [1,2,3,4,5]. The construction waste recycling process is generally applied to materials such as asphalt, brick, concrete, ferrous metal, and glass [6]. However, construction demolition waste such as concrete and brick, is usually dumped into the ground without being reused [7]. In this case, the ecological environment has deteriorated while the lands are occupied with debris. These wastes are sometimes used in non-structural components and road applications [8,9]. Brick, one of the construction demolition waste materials, is generally evaluated in this way. Although brick waste is generally used as filling and stabilizing material, researchers have studied its use in regular concrete, alkali-activated/geopolymer composites, and other waste materials [10,11,12,13,14]. From the data literature, it has been seen that the addition of 10–20% of brick waste instead of cement increases the compressive [15,16,17]. Arif et al. [18] used brick powder waste in concrete mixtures by replacing cement at 5% and 10%. In the case of the 10% replacement of brick powder, the 28-day compressive, flexural, and splitting tensile strengths were found to increase by 10, 24, and 12%, respectively, compared to the control mixtures.

Concrete plays the most crucial role in the construction industry. However, in order to maintain this vital role, it must also be environmentally friendly. This is due to how in concrete production, the carbon footprint of concrete emerges at every stage, from obtaining raw materials to transporting concrete to placing the concrete. For example, in the production process of cement, one of the concrete raw materials, approximately 850 kg of CO_2_ is released in nature to obtain one ton of clinker [19]. In order to reduce this harmful effect, supplementary cementitious materials (SCMs) are used instead of cement. SCMs conserve natural resources and assist in waste management by incorporating urban and industrial waste by-products into concrete. In addition, one of the essential topics researched today is the production of alkali-activated/geopolymer composites. Alkali-activated/geopolymer composites represent one of the first efforts in the search for greener concrete. Due to the integration of high volumes of industrial waste materials (by-products) such as fly ash, waste glass powder, blast furnace slag, and silica fume, alkali-activated/geopolymer composites are seen as a way to significantly reduce the environmental carbon footprint of concrete production in terms of method and energy and CO_2_ emissions [20].

Alkali-activated/geopolymer composites are produced by activating aluminosilicate-based materials with various alkali activators. These activators are usually sodium silicate (Na_2_SiO_3_), sodium hydroxide (NaOH), and potassium hydroxide (KOH), which are sometimes used together or individually [21,22,23]. Although KOH has higher alkalinity than NaOH, it has a lower activation potential. Altan and Erdoğan [24] showed that when the samples activated with NaOH were compared with those with KOH, the samples produced with NaOH reached higher strengths after the first week. The other main components of alkali-activated/geopolymer composites are aluminum silicate (binder), sand, and aggregates [25]. The properties of alkali-activated/geopolymer composites are influenced by the chemical content of the industrial waste materials used. For instance, industrial waste materials containing CaO can react with water and form calcium silicate hydrated compounds that enhance the mechanical properties of alkali-activated/geopolymer composites [26]. Different gel structures can be formed depending on the calcium content. For instance, low calcium content can lead to the formation of N-A-S-H (sodium alumina silicate hydrate) gels, while high calcium content can lead to the formation of C-A-S-H (calcium alumina silicate hydrate) gels. Geopolymers are a type of alkali-activated materials that are produced by activating a precursor using an alkaline solution under an appropriate temperature environment. Geopolymerization occurs through the formation of polysialic 3D networks of monomers consisting of SiO_4_ and AlO_4_ tetrahedrons. Since industrial waste materials are used to produce alkali-activated/geopolymer composites, the heat of hydration in the reaction process is lower compared to Portland cement. In addition, alkali-activated/geopolymer composites are utilized in various applications due to their high compressive strength and high acid resistance [27,28].

Today, researchers are investigating the performance of alkali-activated/geopolymer composites under the elevated temperature effect. Zhang et al. [29] found that the compressive strength of alkali-activated/geopolymer composites decreased less than conventional Portland cement mortars when exposed to elevated temperatures. Sivasakthi et al. [30] stated that the dimensional stability of alkali-activated/geopolymer composites remained unchanged for up to 800 °C. Kljajevic et al. [31] determined that the thermal effect at 900 °C caused significant morphological changes in alkali-activated/geopolymer composites, leading to a reduction in oxygen and sodium content and the formation of a complex pore structure. Lahoti et al. [32] found that the compressive strength of all alkali-activated/geopolymer composites decreased after exposure to elevated temperatures. Insignificant cracks were detected on the surface of alkali-activated/geopolymer composites produced with conventional Portland cement and exposed to temperatures of 800–1000 °C. When the alkali-activated/geopolymer composites were exposed to 800 °C, the beam with a 20 mm concrete cover could carry a load of approximately 66% of its compressive strength. The beam with a 40 mm coating showed resistance for up to 75% of the load [33]. These studies have determined that the type of alkali activator used in producing alkali-activated/geopolymer composites is also effective in resisting elevated temperatures. Compared with alkali-activated/geopolymer composites containing potassium activators, those produced with sodium activators showed high compressive strength at ambient temperatures and improved compressive strength at elevated temperatures for up to 400 °C. The compressive strength of alkali-activated/geopolymer composites produced with a potassium activator at 600 °C is slightly higher than that of those containing sodium [34].

As for the geopolymer composites, brick, which is considered an alumina-silica source [7], is fired at elevated temperatures during the production process, so its performance under the influence of elevated temperature in cement mortars, concrete, and geopolymer composites with brick powder substitutes is also being investigated. The production of brick between 850–950 °C ensures that the brick is resistant to elevated temperatures. When cementitious composites are exposed to elevated temperatures, their strength decreases significantly [35,36,37]. The compressive strength of concrete depends on the stability of the C-S-H structure. When concrete is exposed to temperatures of 500 °C and above, some gel structures turn into crystalline particles, and the volume of the capillary pores in the concrete increases [29]. This situation adversely affects the compressive strength of concrete. When geopolymer composites are exposed to elevated temperatures such as 800 °C, the compressive strength of geopolymer composites decreases due to the thermal incompatibility between the geopolymer composites and aggregate [38]. Many studies in the literature have stated that fly ash-based-alkali-activated composites cured at 60 °C and 80 °C exhibit high early strength and excellent fire resistance [39,40,41].

Some parameters influence the effect of elevated temperatures, such as the sample cooling process. There are various types of cooling, such as air and water. Kara and Arslan [42] investigated the high-temperature effect of plasticizer and antifreeze additives on cementitious composites. They found that the decrease in strength was the highest in the mixtures where plasticizer and antifreeze were used together for both cooling types at temperatures of 550 °C and 700 °C. In contrast, the lowest strength decrease was obtained from the samples with antifreeze added to the water cooler. Pan et al. [43] found that sodium alumina silicate hydrate (N-A-S-H) gel is rich in calcium in alkali-activated composites.

After conducting a thorough literature review, it has been observed that there are gaps in understanding the effects of FBP used as a substitute for slag in SGC and the impact of cooling processes in air and water when exposed to elevated temperatures. Therefore, this study aims to investigate the effect of elevated temperatures on SGC with the inclusion of FBP. In this regard, in the preliminary trials performed within the scope of the study, optimal conditions for geopolymer composites were determined as 16 molarity, 100 °C, and 24 h of curing time. Then, FBP from CDW was substituted in different replacement ratios (10%, 20%, 30%, and 40% by weight of slag) into the SGCs with optimum molarity, curing temperature, and curing time. The produced SGCs were exposed to elevated temperature effects at 300 °C, 600 °C, and 800 °C and then subjected to air- and water-cooling regimes. The ultrasonic pulse velocity (U_pv_), flexural strength (f_fs_), compressive strength (f_cs_), and mass loss of the SGCs with the inclusion of FBP were determined. Additionally, scanning electron microscopy (SEM) analyses were carried out for control, and 20% FBP-based SGCs were cooled in air and water after elevated temperatures of 300 °C and 600 °C. Finally, the results were compared with the samples tested in the laboratory environment (20 °C), and the performance of FBP-based SGCs under elevated temperatures was determined.

## 2. Experimental Study

### 2.1. Materials

Standard sand complying with EN 196-1 [44] was used to prepare the mortar mixes. Ground granulated blast furnace slag (BFS) was procured from the Zonguldak Eregli iron and steel plant in Turkey. Firebrick powder (FBP) was obtained by grinding waste firebricks from construction demolition waste (CDW). FBP was ground according to the order in Figure 1 and used in geopolymer composite mortar mixtures after being sieved through a 75 µm sieve (No. 200) in the final stage. The CaO ratio of GBFS is 36.09%. However, there is a widespread consensus that a precursor including more than 20% CaO is not promising for polymerization owing to its rapid setting. Hence, FBP was used to replace up to 40% of slag. Note that the CaO ratio of FBP is 0.70%. The physical and chemical components of BFS and FBP, as determined by X-ray fluorescence spectrometer (XRF) analysis, are given in Table 1. A Rigaku ZSX Primus II device was used for XRF testing. Solid NaOH was used as an alkaline activator to activate the mixture of BFS and BFS + FBP in this study. The NaOH used in the study is 99% pure and had a molecular weight of 40 g/mol. Potable tap water was used to prepare the NaOH solution. NaOH has shown a more increased ability to dissolve aluminate and silicate, making it suitable for composing alkaline solutions with sodium silicate. The use of sodium silicate can enhance the mechanical properties of geopolymers. However, it is not cost-effective, and a significant amount of energy is required during its production.

### 2.2. Preparation of the Geopolymer Composite Mortar Mixtures

Based on the literature review, it has been seen that there is a correlation between the compressive strength of geopolymer composites and the curing temperature [45,46]. Villa et al. [47] suggested that heat curing should be conducted at temperatures such as 60–80 °C to achieve rapid compressive strength development. However, in their study, Guzelkucuk and Demir [48] obtained the highest compressive strength results in geopolymer composites with 24-h heat curing at 110 °C. In order to determine the optimum molarity, curing temperature, and curing time for the geopolymer composites in this study, preliminary trials were conducted under the conditions shown in Figure 2, taking into account the studies in the literature.

After the preliminary trials with 100% slag-based-geopolymers, the optimum molarity, curing temperature, and curing times that achieved the highest compressive strength were determined to be 16 M, 100 °C, and 24 h, respectively, as given in Table 2. These parameters were kept constant for the second stage of the study.

After determining optimum molarity, curing time, and curing temperature, five different batches were prepared by replacing BFS with FBP at up to 40% replacement ratios. The mixture proportions of the different geopolymer composite mortars are given in Table 3. The water-to-binder ratio was kept constant in the mixture proportions. The total amount of binder in the mixtures was 450 g, and the FBP was replaced by 10%, 20%, 30%, and 40% of the slag weight. After preparing the geopolymer composite mortar mixtures, prismatic samples measuring 40 × 40 × 160 mm were produced by curing them in an oven at 100 °C for 24 h. The production steps of the samples are summarized in Figure 3.

### 2.3. Elevated Temperatures and Different Cooling Regimes

For each batch, six geopolymer composite mortar samples were taken out of the oven after 24 h and subjected to 300 °C, 600 °C, and 800 °C. The furnace heating rate was set at 6 °C/min, and the geopolymer samples were exposed to the target temperatures for 3 h. After that, the furnace was turned off, and half of the samples were air-cooled while the other half were water-cooled. The samples that were water-cooled were placed in buckets filled with water. Air cooling was achieved by leaving the samples at laboratory conditions until they reached an ambient temperature, with a temperature drop of approximately 1.1 °C/min. The heating regime used in the study is presented in Figure 4, while the processing steps of the elevated temperature application and different cooling regimes on the samples are summarized in Figure 5.

### 2.4. Ultrasonic Pulse Velocity

An ultrasonic pulse velocity (U_pv_) test was conducted to measure the quality of geopolymer composite mortars. The measurements were carried out on geopolymer samples that were exposed to different elevated temperatures (300 °C, 600 °C, and 800 °C), followed by air- and water-cooling regimes. The U_pv_ measurements were taken when the geopolymer samples cooled in the air and reached the laboratory temperature. The U_pv_ measurement of the geopolymer samples cooled in water was conducted after drying in an oven at 105 ± 5 °C for 24 h. The U_pv_ tests of the samples were performed with an accuracy of 0.10 s in accordance with ASTM C 597-16 [49] standard. Table 4 shows the quality of concrete as a function of the ultrasonic pulse velocity speed [50,51]. The ultrasonic pulse velocity test of the geopolymer composites was conducted using the Proceq Pundit Lab+.

### 2.5. Flexural Strength

The flexural strength (f_fs_) test was conducted according to EN 196-1 [44]. The test was performed on geopolymer samples exposed to different elevated temperatures (300 °C, 600 °C, and 800 °C) and then exposed to air- and water-cooling regimes. The f_fs_ was carried out on prismatic geopolymer mortars with dimensions of 40 × 40 × 160 mm, and the average f_fs_ of three samples was taken as the final flexural strength result. The f_fs_ test was performed under three-point loading with a loading rate of 50 ± 10 N/s, using a UTCM-6431 coded device from a UTEST company in the experiments.

### 2.6. Compressive Strength

Compressive strengths (f_cs_) were determined in accordance with EN 196-1 [44]. The f_cs_ test was performed on six broken pieces of prismatic geopolymer samples from 3-point flexural tests. The average f_cs_ of these six samples was taken as the final f_cs_ result. Geopolymer composite samples were loaded from a cross-sectional area of 40 × 40 mm at a loading rate of 2400  ±  200 N/s. The experiments were conducted using a UTCM-6431 coded device belonging to the UTEST.

### 2.7. Mass Loss

The mass of the geopolymers was measured in the laboratory before being exposed to high temperatures of 300 °C, 600 °C, and 800 °C. After cooling to laboratory conditions, their masses were measured again. The mass loss of geopolymers cooled in the air after exposure to high temperatures was also measured after reaching laboratory conditions. Mass loss measurement of geopolymers cooled in water was performed after drying in an oven at 105 ± 5 °C for 24 h. The mass loss measured after exposure to high temperatures and different cooling conditions was compared with the measurements performed in the laboratory and the final mass results were determined.

### 2.8. Microstructural Analyses

SEM analyses were conducted for the geopolymer composite samples with 0% (Ref), and 20% FBP substituted geopolymer mixtures (20FBP), which were exposed to different elevated temperatures (300 °C, 600 °C) and then subjected to air- and water-cooling regimes. Microstructure analyses of the geopolymers were carried out using a Zeiss EVO 40XP SEM instrument. The analyses were performed on small geopolymer samples obtained after the compressive strength. To obtain clear images in the SEM analysis, Small geopolymers were coated with gold.

## 3. Results and Discussion

### 3.1. Ultrasonic Pulse Velocity

#### 3.1.1. Ultrasonic Pulse Velocity Findings of Air-Cooled Geopolymer Composite Samples after Elevated Temperature

Figure 6 shows the results of the ultrasonic pulse velocity (U_pv_) test performed on geopolymers cooled in the air after exposure to elevated temperatures, and relative residual (U_w_) results compared to samples tested at 20 °C. Figure 6 illustrates that at 20 °C, the U_pv_ results vary between 3.89 and 4.83 km/s, with the U_pv_ results decreasing as the FBP replacement ratio increases. When the geopolymer composite samples were exposed to 300 °C and cooled in air, decreases in U_pv_ results were observed. The U_pv_ results for the geopolymer composite samples with Ref, and the 10FBP, 20FBP, 30FBP, and 40FBP samples were determined to be 3.76, 3.96, 4.19, 3.92, and 3.69 km/s, respectively. The relative residual results of these geopolymer composite samples were determined to be 77.82, 84.05, 90.20, 93.22, and 94.86%, respectively. It is observed that the decreases in U_pv_ results become more pronounced with an increasing temperature above 300 °C. The relative residual results of the geopolymer composite samples exposed to temperatures of 600 °C and 800 °C varied between 53.00–61.47%, and 18.10–26.49%, respectively. According to these results, the decrease in the U_pv_ of geopolymer composite samples decreased with the increase in the FBP replacement ratio. Especially at higher temperatures, the decreases in the U_pv_ were more pronounced. The decrease in U_pv_ results with the increase in temperature was attributed to the increase in the porosity of the geopolymer composite samples. The free water and OH groups of alkaline geopolymer composite mortars evaporate at around 150 °C [52,53]. It is believed that this situation causes decreases in U_pv_ results, as it increases the porosity of the geopolymer composite mortars. According to the Whitehurst [50] classification, the geopolymer composite mortars were categorized as good, poor, and very poor categories after exposure to temperatures of 300 °C, 600 °C, and 800 °C, respectively.

#### 3.1.2. Ultrasonic Pulse Velocity Findings of Water-Cooled Geopolymer Composite Samples after Elevated Temperature

Figure 7 shows the U_pv_ test of geopolymers cooled in the water after elevated temperature as well as U_w_ results compared to samples tested at 20 °C. Figure 7 demonstrates that the U_pv_ results of the geopolymers cooled in water after temperatures of 20 °C and 300 °C decrease with the increase in the FBP replacement ratio. When examining the U_pv_ results of the geopolymer samples cooled in water after exposure to 600 °C, it is seen that the U_pv_ results of the geopolymer composite mortars with 10% and 20% FBP substitution are higher than those of the Ref mortars. The relative residual results of the geopolymer samples exposed to temperatures of 300 °C and 600 °C vary between 67.94–72.82% and 32.79–38.56%, respectively. The U_pv_ measurement could not be performed because Ref, 10FBP, and 20FBP geopolymer mortars cooled in water after exposure to a temperature of 800 °C lost their cross-section area. This decrease in U_pv_ results may be caused by the sudden cooling with water. As a result of the measurement conducted on the 30FBP and 40FBP geopolymer mortars, the relative residual results were found to be 16.85% and 17.46%, respectively. The porosity of geopolymer composite mortars increases with the deterioration of the C-S-H gel after exposure to 600 °C, and this deterioration increases with sudden cooling [19,53,54]. As a result, significant decreases were observed in U_pv_ results. The U_pv_ results of the geopolymer samples exposed to temperatures of 300 °C, 600 °C, and 800 °C vary 2.75–3.52 km/s, 1.46–1.82 km/s, and 0.71–0.68 km/s, respectively. According to the Whitehurst [50] classification, after the geopolymer mortar samples were exposed to temperatures of 300 °C, 600 °C and 800 °C and cooled in water, they were placed in the “doubtful”, poor”, “very poor”, and “very poor” categories, respectively.

#### 3.1.3. Comparison of Ultrasonic Pulse Velocity Results of Geopolymer Composite Samples Cooled in Air and Water after Elevated Temperature

The U_pv_ results of the geopolymer samples cooled in air and water after elevated temperature are shown in Figure 8 and the effect of cooling regimes on the U_pv_ results of the geopolymer samples can be seen more clearly.

Figure 8 illustrates that the Upv results of the geopolymers cooled in water were lower than those cooled in air. This decrease is particularly evident in geopolymers exposed to temperatures of 600 °C and 800 °C. This may be attributed to the disruption of C-S-H when the samples are suddenly cooled after being exposed to high temperatures, leading to an increase in sample porosity and lower U_pv_ results. When comparing U_pv_ results of Ref, 10FBP, 20FBP, 30FBP, and 40FBP samples that were cooled in air and water after being exposed to 300 °C, decreases of 6.43%, 14.18%, 24.68%, 23.54%, and 25.50%, respectively, were observed. Therefore, it can be concluded that the U_pv_ decrease in Ref samples exposed to a temperature of 300 °C is relatively low. It was determined that the decreases in U_pv_ results of Ref, 10FBP, 20FBP, 30FBP, and 40FBP samples exposed to a temperature of 600 °C were 38.13%, 33.87%, 36.31%, 39.47%, and 38.90%, respectively. The decreases in U_pv_ results of 30FBP and 40FBP samples exposed to a temperature of 800 °C were 36.39% and 31.76%, respectively. The decrease in Upv results of Ref samples exposed to a temperature of 600 °C was higher than that of samples with 10% and 20% FBP replacement. The U_pv_ measurement could not be taken from Ref, 10FBP, and 20FBP samples exposed to a temperature of 800 °C after cooling in water. It was determined that the decreases in the U_pv_ results of the 30FBP and 40FBP samples were 36.39% and 31.76%, respectively. Figure 9 illustrates the surface appearances of the geopolymers after air-cooling. The images show that the cracks intensified on the surface of the geopolymers and the porosity increased with temperature, particularly at the temperatures of 600 °C and above. These surface changes confirm the U_pv_ test results. Figure 10 shows the surface appearances of the geopolymers after the water-cooling. It was observed that the porosity on the surface of the samples increased with the increase in temperature, as in the samples cooled in air. In addition, fragmentation was observed on the surface of Ref, 10FBP, and 20FBP samples cooled in water after exposure to a temperature of 800 °C. When the surface appearances of these samples are examined, aggregate particles in the samples were seen. Figure 9 and Figure 10 demonstrate that the water-cooling process significantly affected the color change of the samples.

### 3.2. Flexural Strength

#### 3.2.1. Air-Cooled Geopolymers after Elevated Temperatures

Figure 11 illustrates the flexural strength (f_fs_) results of geopolymers cooled in the air after high temperature, as well as the relative residual results compared to samples tested at 20 °C.

When Figure 11 is examined, it can be seen that the f_fs_ results of the geopolymer composite samples at 20 °C ranged from 11.38 MPa and 7.61 MPa, and the f_fs_ results decreased with the increase in the FBP replacement ratio. Significant decreases were observed in the flexural strength of the geopolymer composite samples after exposure to elevated temperature, which is consistent with previous studies in the literature [55]. After exposure to 300 °C and air cooling, the f_fs_ results for Ref, 10FBP, 20FBP, 30FBP, and 40FBP samples were determined to be 7.27, 7.89, 8.19, 7.75, and 7.06 MPa, respectively. The relative residual results of these samples were determined to be 63.88%, 72.62%, 82.15%, 91.33%, and 92.77%, respectively. The f_fs_ results of the geopolymer composite samples exposed to a temperature of 600 °C vary from 3.69 MPa to 4.32 MPa, while the relative residual results of these samples are between 33.04 and 48.49%. While the f_fs_ results of the geopolymer composite samples exposed to a temperature of 800 °C after air cooling were between 1.98 and 2.63 MPa, their relative residual results varied between 17.40 and 32.46%. When examining the relative residual results of the geopolymer composite samples cooled in the air after exposure to temperatures of 300 °C, 600 °C, and 800 °C, it is obvious that the strength losses decrease with the increase of the FBP replacement ratio. For all temperature values, the minor strength loss was observed in the 40% FBP substituted samples, while the highest strength loss was observed in the samples without FBP replacement. When the results were evaluated, the performance of geopolymer composite mortars against elevated temperatures could be improved by replacing FBP with blast furnace slag. The higher f_fs_ results of FBP-substituted samples at elevated temperatures compared to Ref samples are thought to be due to the increased geopolymerization and higher FBP activation in the binder matrix. Celikten et al. [53] obtained similar results for calcined perlite-substituted geopolymer mortars in their study. Therefore, the increase in f_fs_ results at elevated temperatures was attributed to the sintering reactions of unreacted FBP particles.

The decrease in flexural strength of geopolymer composite samples after reaching 300 °C was explained as a result of the evaporation of water in the voids of the matrix [55,56]. Fares et al. [57] observed that the bonds between water and hydration products start to weaken at around 200 °C. When the geopolymer composite samples are exposed to a temperature range of 400–600 °C, free and chemically bound water is expelled. For this reason, it has been stated that the strength loss in samples is caused by chemical transformations in the hydration products, including the C–S–H and C-A-S-H components [55]. When the temperature rises above 600 °C, excessive shrinkage occurs as a result of moisture loss, leading to a significant deterioration in the microstructure of the geopolymer composite mortar. In addition, when the temperature exceeds 600 °C, the strength of the C-A-S-H gel also decreases significantly [58].

#### 3.2.2. Water-Cooled Geopolymers after Elevated Temperature

Figure 12 depicts the f_fs_ results of geopolymers cooled in the water after high temperature and the relative residual results compared to samples tested at 20 °C.

Figure 12 demonstrates that there were decreases in flexural strength results observed when the geopolymer composite samples were exposed to elevated temperatures and subsequently cooled in water. The flexural strength results for Ref, 10FBP, 20FBP, 30FBP, and 40FBP samples cooled in water after 300 °C were determined to be 6.94, 6.71, 6.30, 5.66, and 5.12 MPa, respectively. Relative residual results of these samples were determined as 60.98%, 61.76%, 63.19%, 66.70%, and 67.28%, respectively. The relative residual results at 300 °C show that the strength loss of Ref samples is higher than FBP substituted samples. The flexural strength results of the geopolymer composite samples exposed to 600 °C vary from 2.79 MPa to 3.53 MPa. Relative residual results of these samples vary between 27.24 and 36.66%, while the highest loss of compressive strength at 600 °C was observed in Ref samples. Ref, 10FBP, and 20FBP samples cooled in water after 800 °C were not subjected to flexural strength test since they lost cross-section. This may be caused by their sudden cooling with water. The flexural strength results of the 30FBP and 40FBP samples were measured as 1.63 MPa and 1.47 MPa, respectively. Relative residual results were determined to be 19.21% and 19.32%, respectively. At temperatures above 300 °C for water-cooled samples, the importance of FBP substitution was more evident. When the results are evaluated, it has been determined that FBP can be replaced with BFS for up to 30%.

#### 3.2.3. Comparison of Flexural Strength Results of Geopolymers Cooled in Air and Water after Elevated Temperature

The flexural strength results of the geopolymer composite samples cooled in air and water after elevated temperature are shown in Figure 13 to illustrate the effect of cooling regimes on the flexural strength of the geopolymer composite samples.

As seen in Figure 13, the flexural strength results of the geopolymer composite samples cooled in water were lower than those cooled in air. The decrease in flexural strength results of geopolymer composite samples cooled in water after being exposed to 300 °C compared to geopolymer composite samples cooled in the air was determined to be 4.54%, 14.96%, 26.37%, 25.68%, and 27.48% for the Ref, 10FBP, 20FBP, 30FBP, and 40FBP samples, respectively. The decrease in flexural strength of geopolymer composite samples cooled in water after exposure to 600 °C, compared to samples cooled in air, was determined to be 17.55%, 14.11%, 22.22%, 26.88%, and 24.39% for the Ref, 10FBP, 20FBP, 30FBP, and 40FBP samples, respectively. The Ref, 10FBP, and 20FBP samples were cooled in water after reaching 800 °C and experienced lost cross-section. Therefore, flexural strengths were not measured. The decrease in flexural strength of 30FBP and 40FBP samples was determined to be 38.02% and 40.49%, respectively. When the results were evaluated, it was concluded that water cooling the geopolymer samples exposed to elevated temperatures was not suitable for materials exposed to longer periods or higher temperatures. For this reason, the suggestion that alternative cooling methods should be used instead of water for cementitious composites was also valid for geopolymer mortars [59].

### 3.3. Compressive Strength

#### 3.3.1. Air-Cooled Geopolymers after Elevated Temperature

Figure 14 demonstrates the compressive strength results of geopolymers cooled in the air after a high temperature and the relative residual results compared to samples tested at 20 °C.

Figure 14 illustrates that the compressive strength results of the geopolymer composite samples at 20 °C ranged from 69.24 MPa and 40.80 Mpa, with a decrease observed with increasing the FBP replacement ratio. After exposure to 300 °C and air cooling, the compressive strength results of samples decreased. The compressive strength of the Ref, 10FBP, 20FBP, 30FBP, and 40FBP samples was determined to be 38.29, 41.46, 43.47, 40.85, and 36.33 MPa, respectively. The relative residual results of these samples were determined to be 54.92%, 66.80%, 71.51%, 80.05%, and 89.03%, respectively. When examining the relative residual results at 300 °C, it can be observed that the compressive strength decreases as the FBP replacement ratio increases. Geopolymer composite samples with 40% of FBP replacement showed a minor decrease in compressive strength. The decrease in strength at 300 °C is attributed to the excessive vapor pressure caused by the evaporating free water in the geopolymer composite samples. As a result, significant cracks can be observed in the geopolymer composite samples [56]. The compressive strength results of the geopolymer composite samples exposed to a temperature of 600 °C range from 18.22 MPa to 21.36 MPa. The relative residual results of these samples range from 27.56 to 44.66%. As the temperatures increase from 400 to 600 °C, the strength decrease intensifies due to the degradation of chemical bonds, leading to the deterioration of the microstructure. Additionally, it has been reported that high-temperature exposure to air can create voids and cracks in the interfacial zone between the aggregate and matrix [55,60]. At 800 °C, the compressive strength results range from 8.22 to 12.39 Mpa, and the relative residual results range from 11.79 to 27.42%. The highest decrease in compressive strength after exposure to 600 °C and 800 °C was observed in the Ref samples. When evaluating the results, it was found that the performance of geopolymer composite mortars against elevated temperatures could be improved by replacing the FBP with blast furnace slag. Çelikten et al. [53] determined that the strength of calcined perlite-based geopolymer mortars after exposure to the elevated temperature was higher than that of the Ref mortars. This increase was attributed to the sintering reactions of unreacted calcined perlite particles. A similar situation was observed in this study, and the higher strength of the FBP-substituted geopolymer composite mortars after exposure to elevated temperature than the Ref mortars was also attributed to the sintering reactions of the FBP particles. The decrease in compressive strength of geopolymer composite mortars with increasing temperature was found to be consistent with the literature [61,62].

#### 3.3.2. Water-Cooled Geopolymers after Elevated Temperature

Figure 15 demonstrates the compressive strength results of geopolymers cooled in water after exposure to high temperatures and the relative residual results compared to samples tested at 20 °C.

When the geopolymers are exposed to high temperatures and subsequently cooled in water, a substantial decrease in strength is observed. The compressive strength results of samples cooled in water after being subjected to 300 °C were 34.05, 31.06, 30.89, 26.42, and 21.50 MPa for the Ref, 10FBP, 20FBP, 30FBP, and 40FBP samples, respectively. The relative residual results were determined to be 48.84, 50.05, 50.82, 51.77, and 52.70%, for the Ref, 10FBP, 20FBP, 30FBP, and 40FBP samples, respectively. It was found that the decrease in strength of the Ref sample was higher than that of the FBP-based geopolymer samples, as evidenced by the relative residual results at 300 °C. The results of the geopolymers exposed to 600 °C varied between 13.60 MPa and 16.30 Mpa, with relative residual results ranging from 22.56 to 33.34%. The highest decrease in strength at 600 °C was observed in Ref. The Ref, 10FBP, and 20FBP geopolymer samples cooled in water after 800 °C were not subjected to the compressive strength test because they failed cross-sections, possibly due to their sudden cooling with water. The results of the 30FBP and 40FBP samples were determined as 6.57 and 5.40 MPa, with relative residual results of 12.87 and 13.23%. These results suggest that the FBP could be replaced with BFS for up to 30%. The contribution of the FBP to the compressive strength results of geopolymer composite mortars was more evident with an increase in both the FBP replacement ratio and temperature.

#### 3.3.3. Comparison of Results of Geopolymers Cooled in Air and Water after Elevated Temperature

The compressive strength results of the geopolymer composite samples cooled in air and water after elevated temperature are shown in Figure 16 so that the effect of cooling regimes on the compressive strength of the geopolymer composite samples can be seen more clearly.

As seen in Figure 16, the compressive strength results of the geopolymer composite samples cooled in water were lower than those cooled in air. This decrease is more clearly seen especially in samples exposed to 800 °C. The decrease in compressive strength of the geopolymer composite samples cooled in water after exposure to 300 °C compared to the samples cooled in the air was determined to be 11.07%, 24.83%, 28.93%, 35.33%, and 40.81% for the Ref, 10FBP, 20FBP, 30FBP, and 40FBP samples, respectively. The decrease in compressive strength of geopolymer composite samples cooled in water after exposure to 600 °C, compared to samples cooled in air, was determined to be 18.13%, 23.32%, 25.11%, 29.40%, and 25.34% for the Ref, 10FBP, 20FBP, 30FBP, and 40FBP samples, respectively. Samples of Ref, 10FBP, and 20FBP were cooled in water after exposure to 800 °C and they had their cross-sections damaged. Therefore, compressive strength results were not measured for these samples. The decrease in compressive strength in 30FBP and 40FBP samples was determined to be 46.99% and 51.77%, respectively. Water cooling was found to be unsuitable for geopolymer composite mortars that were exposed to elevated temperatures. Therefore, it is recommended that alternative cooling methods should be investigated instead for use with cementitious composites instead of water.

Figure 17 shows the images of 30FBP and 40FBP samples after being cooled in air and water following exposure to a temperature of 800 °C, and then subjected to a compressive strength test. Upon examination of Figure 17, it is evident that the voids in the mortars cooled with water are more tightly packed. In contrast, the voids in the air-cooled mortars are less pronounced and dispersed over a smaller area. Thus, the negative impact of water-cooling on geopolymer mortars after exposure to high temperatures is apparent. This section concluded that alternative cooling methods should be explored instead of water-cooling for geopolymer mortars after exposure to high temperatures. Further research is needed in this area.

### 3.4. Mass Loss

#### 3.4.1. Mass Loss Findings of Air-Cooled Geopolymer Composite Samples after Elevated Temperature

Figure 18 shows the mass loss results of the geopolymer composite samples after exposure to elevated temperatures, where the samples were cooled in air. The measurements were taken before and after the samples were subjected to heat.

Upon examining the mass loss results of the geopolymer composite samples cooled in the air after being exposed to elevated temperatures, it was observed that the mass loss results increased in all mixture groups as the temperature increased. The loss of mass in the samples following exposure to elevated temperature is regarded as an indication of the evaporation of the water in the mortar and the deterioration of the pore structure [55]. For the Ref samples, the mass loss results following exposure to temperatures of 300 °C, 600 °C, and 900 °C were determined to be 4.61%, 6.65%, and 11.33%, respectively. The mass loss results were determined as 4.47%, 4.32%, 4.21%, and 3.98% for 10FBP, 20FBP, 30FBP, and 40FBP samples exposed to 300 °C temperature, respectively. With the FBP replacement ratio increasing to 40%, a 13.67% reduction in mass loss results was achieved compared to the Ref samples. Mass loss results of Ref, 10FBP, 20FBP, 30FBP, and 40FBP samples exposed to 600 °C temperature were determined to be 6.65%, 6.32%, 6.16%, 5.96%, and 5.79%, respectively. With the FBP replacement ratio increasing to 40%, a 12.93% reduction in mass loss results was achieved compared to the Ref samples. The mass loss results of the Ref, 10FBP, 20FBP, 30FBP, and 40FBP samples exposed to 800 °C temperature were determined to be 11.33%, 11.09%, 10.93%, 10.61%, and 10.32%, respectively. With the FBP replacement ratio increasing to 40%, an 8.91% reduction in mass loss results was achieved compared to the Ref samples. At least 80% of the mass loss of alkali-activated/geopolymer binders is within the first 200 °C [54,63]. This loss is attributed to the evaporation of free water and OH groups presented in the pores of the mortar matrix [52,53]. Most of the mass loss between 200 and 600 °C is due to how possible degradation of the C–S–H gel significantly affects the loss of strength [19,54]. The disruption of the C–S–H gel also dramatically influences the loss of strength [53]. At 600 and 800 °C, the microstructure further deteriorates due to the breakdown of Si–O–Al bonds in the calcium alumina-silicate hydrate (C-A-S-H) gel [55,58]. Upon examining the results, it was determined that the mass loss decreased with the increase of the FBP replacement ratio.

#### 3.4.2. Mass Loss Findings of Water-Cooled Geopolymer Composite Samples after Elevated Temperature

The mass loss results of the geopolymer composite samples, which were cooled in water after being subjected to elevated temperatures, were measured before and after exposure to heat. These results are presented in Figure 19.

Figure 19 shows that the mass loss results of all groups increased with the temperature. The results of 4.27, 6.03, and 12.23% were obtained for the Ref exposed to 300, 600, and 800 °C, respectively. The results decreased with the increase of the FBP. The results of 10FBP, 20FBP, 30FBP, and 40FBP samples exposed to 300 °C were found to be 3.39, 3.86, 3.78, and 3.67%, respectively. With the FBP replacement ratio increasing to 40%, a 14.05% reduction in mass loss results was achieved compared to the Ref samples. Mass loss of Ref, 10FBP, 20FBP, 30FBP, and 40FBP samples exposed to 600 °C temperature was determined as 6.03%, 5.89%, 5.73%, 5.51% and 5.32%, respectively. With the FBP replacement ratio increasing to 40%, mass loss results were reduced by 11.77% compared to the Ref samples. The mass loss of the Ref, 10FBP, 20FBP, 30FBP, and 40FBP samples exposed to 800 °C was determined to be 12.23%, 11.96%, 11.75%, 11.34%, and 10.82%, respectively. As there is a cross-section loss in the Ref, 10FBP, and 20FBP samples cooled in water after a temperature of 800 °C, mass loss results were found to be high. The images of the top and side views of Ref, 10FBP, and 20FBP samples are shown in Figure 20 after losing their cross-section. With the FBP replacement ratio increasing to 40%, mass loss results were reduced by 11.53% compared to the Ref samples. When examining the results, it was found that FBP showed a decrease in mass loss results with the increase in temperature, similar to air-cooled mortars.

#### 3.4.3. Comparison of Mass Loss Results of Geopolymer Composite Samples Cooled in Air and Water after Elevated Temperature

Figure 21 indicates that the mass loss results of the geopolymer composite samples cooled in water after exposure to 300 °C and 600 °C are lower than the samples cooled in air. However, due to fragmentation on the surface of the samples cooled in water after 800 °C, the mass loss of these samples was higher compared to those cooled in air.

The decrease in mass loss results of geopolymer composite samples exposed to 300 °C and cooled in water compared to samples cooled in air were determined to be 7.38%, 10.96%, 10.65%, 10.21%, and 7.79% for the Ref, 10FBP, 20FBP, 30FBP, and 40FBP samples, respectively. The decrease in mass loss results of samples cooled in water at 600 °C compared to samples cooled in air is 9.32%, 6.80%, 6.98%, 7.55%, and 8.12% for the Ref, 10FBP, 20FBP, 30FBP, and 40FBP samples, respectively. It is believed that lower the mass loss results of the samples cooled in water after exposure to 300 and 600 °C compared to those cooled in the air are due to the mortars regaining water lost during exposure to high temperatures while cooling with water. The decrease in mass loss results of samples cooled in the air after 800 °C compared to those cooled in water is 7.36%, 7.27%, 6.98%, 6.44%, and 4.62% for the Ref, 10FBP, 20FBP, 30FBP, and 40FBP samples, respectively. As a result, it has been observed that the FBP can increase the resistance of geopolymer composite mortars to the elevated temperature.

### 3.5. Microstructural Analysis

Microstructure examination was performed using scanning electron microscopy (SEM) to support the data of ultrasonic pulse velocity, flexural strength, compressive strength, and mass loss results after elevated temperature presented in the previous sections. Geopolymer composite mortar samples were cut from after compressive strength, whose microstructure analyses were examined by an electron microscope. The images of geopolymer composite samples cooled in air and water, exposed to temperatures of 20 °C, 300 °C, and 600 °C, at 50×, 1000×, 2000× and comparative examinations were conducted at 5000× magnifications.

When examining the SEM images of Ref samples cooled in the air after elevated temperature (100% BFS was used as a binder) at 5000× magnification, it is observed in Figure 22a that the geopolymer bond is quite tight and robust. Here, the gel matrix is formed continuously. In Figure 22b, it is observed that the continuity of the gel-bond structure of the Ref sample is impaired at 300 °C, and the bond structure weakens and separations occur between the aggregate–matrix interfacial zone. As seen in Figure 22c, the continuity of the gel-bond structure of the Ref sample is disrupted. It turns into a porous structure at 600 °C, and the bond matrices in the gel and aggregate structure are weakened by this deterioration. Particularly at 300 °C, the separations between the aggregate and gel can be seen on the surface of the Ref samples.

When examining the SEM images of 20% of FBP samples (80% BFS, 20% FBP used as a binder) cooled in the air after elevated temperatures at 5000× magnification, Figure 23a shows that the geopolymer bond is partially broken compared to the Ref (Figure 22a), and the gel-matrix structure is in a better condition than the Ref sample at 300 °C (Figure 22b), as seen in Figure 23b. Additionally, an unreacted NaOH activator is also visible in Figure 23a. In Figure 23c, it is observed that the continuity of the gel-bond structure is disrupted and the bond matrices in the gel-aggregate structure deteriorate after exposure to 600 °C temperature.

When examining the SEM images of Ref samples cooled in water after elevated temperature at 5000× magnification, it is observed in Figure 24a that the geopolymer bond is quite tight and robust. Here, the gel matrix is formed continuously. In Figure 24b, it is observed that the continuity of the gel-bond structure of the Ref mortars is broken at 300 °C. It is seen that separations occur between the aggregate –matrix interfacial zone. In Figure 24c, it is observed that the continuity of the gel-bond structure of the Ref samples at 600 °C is disrupted and turns into a multi-void structure. The bond matrices in the gel and aggregate structure have deteriorated.

When examining the SEM images of 20% FBP samples cooled in water after elevated temperature (80% BFS 20% BFP used as a binder) at 5000× magnification, it is observed in Figure 25a that concerning the geopolymer bond compared to the Ref gel bond, the structure has partially deteriorated. An unreacted NaOH activator is also seen in Figure 25a. In Figure 25b, the gel-matrix structure showed quite a deterioration. In Figure 25c, on the other hand, the continuity of the gel-bond structure was disrupted due to an elevated temperature (600 °C) and cooling in water. Cracks are visible between the bond matrices in the gel-aggregate structure.

## 4. Conclusions

This study investigated the effect of elevated temperatures on slag-based geopolymer composites with the inclusion of firebrick powder (FBP). In the preliminary trials conducted as part of this study, optimum molarity, curing temperature, and curing time conditions were determined as 16 molarity, 100 °C, and 24 h, respectively. Firebrick powder from construction and demolition waste (CDW) was then substituted in various replacement ratios (10%, 20%, 30%, and 40% by slag weight) into the geopolymer composite mortars with optimum molarity, curing temperature, and curing time. The produced geopolymer mortar samples were exposed to elevated temperatures at 300 °C, 600 °C, and 800 °C and then subjected to air- and water-cooling regimes. The ultrasonic pulse velocity, flexural strength, compressive strength, and mass loss of the slag-based geopolymer composites with the inclusion of FBP were determined. The key outcomes of the study are as follows:Significant decreases in ultrasonic pulse velocity results were observed in the geopolymer composite samples that were exposed to high temperatures. Additionally, it was found that the ultrasonic pulse velocity results of the geopolymers that were cooled in water were lower than those cooled in air. This may be due to the breakdown of the C-S-H phase due to the sudden cooling of the geopolymers after exposure to high temperatures.It is observed that the compressive and flexural strength results of the geopolymer composite samples decrease with the increase in the firebrick powder replacement ratio in the samples tested at 20 °C. When the firebrick powder replacement ratio is used up to 30%, the flexural strength results of the samples are higher than the Ref samples when cooled in the air after temperatures of 300 °C and 600 °C. The Ref samples cooled in the air after a temperature of 800 °C showed the lowest flexural strength result. When firebrick powder was used for up to a 20% replacement ratio, the samples cooled in water exposed to a temperature of 600°C showed a higher flexural strength result than the Ref samples.It was observed that samples cooled in the air after 300 °C and 600 °C had higher compressive strength results for 10%, 20%, and 30% of firebrick powder-substituted samples compared to the Ref samples. However, for samples with 10%, 20%, 30%, and 40% of firebrick powder replacement and cooled air after a temperature of 800 °C, the compressive strength results were higher than the Ref samples. For water-cooled samples exposed to 600 °C, the compressive strength results of 10% and 20% of firebrick powder-substituted samples were higher than the Ref samples. While the highest compressive strength was observed in the Ref samples at all temperatures, the lowest compressive strength was observed in the samples with 40% of firebrick powder replacement.The mass loss results of the geopolymer composite samples, cooled in both air and water after elevated temperature, decreased with the increase of the firebrick powder replacement ratio. This decrease in mass loss was more evident when the temperature value was at 300 °C. A decrease of 13.67% in air-cooled geopolymer composite samples and 14.05% in water-cooled geopolymer composite samples was achieved in mass loss results compared to the Ref samples, with an increase of the firebrick powder replacement rate to 40%.When the geopolymer composite samples cooled in air and water after elevated temperatures were compared, it was observed that the mass loss results of the samples cooled in water after 300 °C and 600 °C were lower than those cooled in air.In the case of water-cooling of the geopolymer composite samples after elevated temperatures, significant fragmentation of the samples was observed, especially at temperatures of 800 °C and above. Therefore, it is suggested that alternative cooling methods should be used instead of water during the cooling of geopolymer composite samples after elevated temperatures. Further research should be conducted on this subject.

The result of the study suggests that the use of firebrick powder in geopolymer composites at a replacement ratio of up to 30% can significantly enhance the elevated temperature resistance.

## Figures and Tables

**Figure 1 polymers-15-02127-f001:**
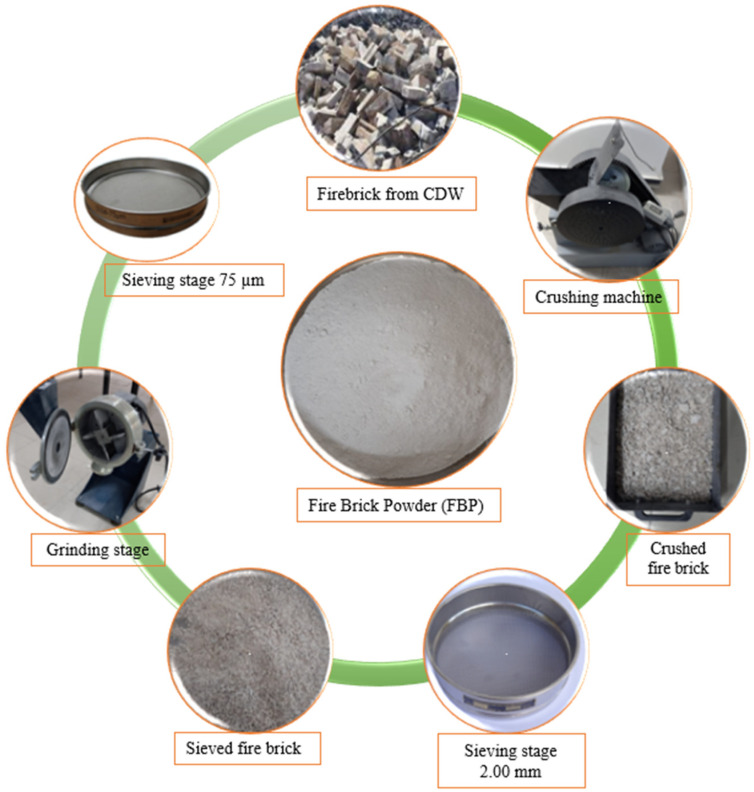
Production stages of firebrick powder.

**Figure 2 polymers-15-02127-f002:**
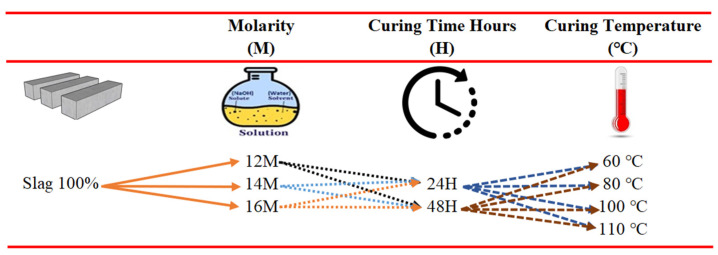
Preliminary trial criteria to determine the optimum molarity, curing time, and temperature.

**Figure 3 polymers-15-02127-f003:**
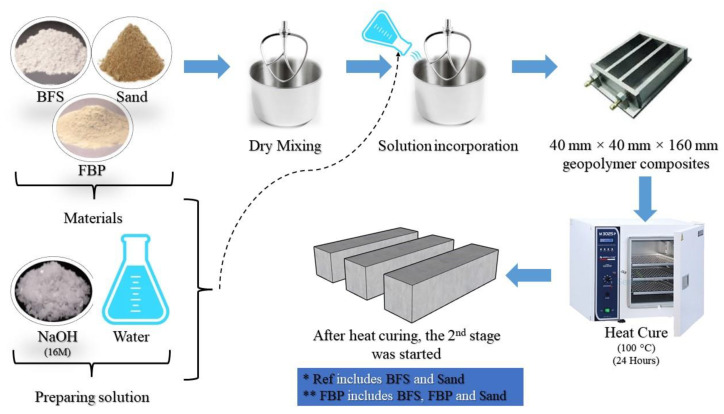
The production steps of the geopolymer composite samples.

**Figure 4 polymers-15-02127-f004:**
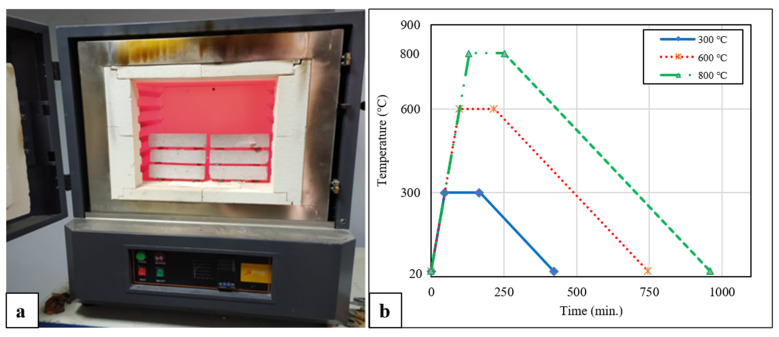
(**a**) Furnace and (**b**) heating regime.

**Figure 5 polymers-15-02127-f005:**
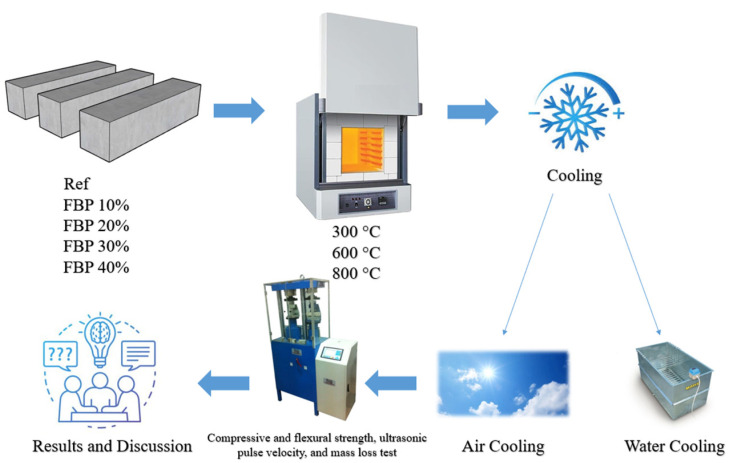
Summary of the processing steps of the elevated temperature application and different cooling regimes.

**Figure 6 polymers-15-02127-f006:**
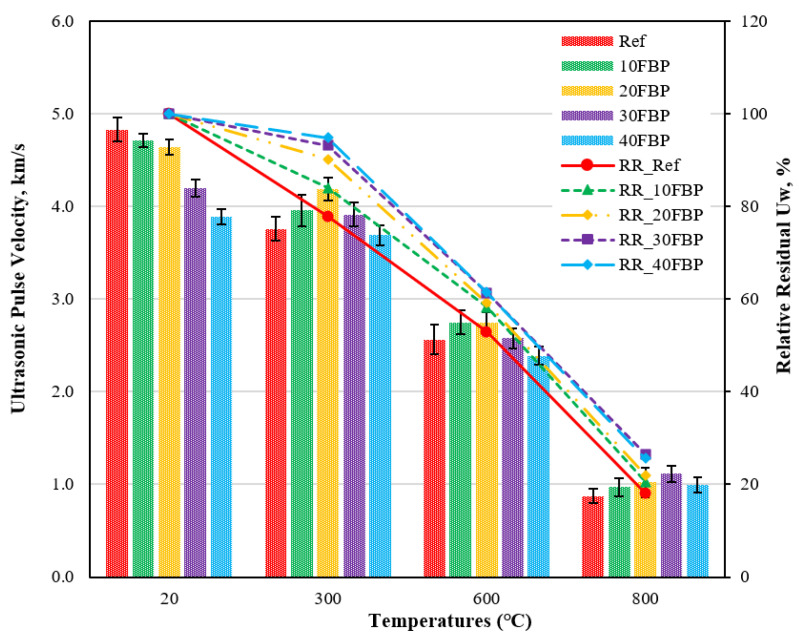
Ultrasonic pulse velocity results of geopolymer composite samples cooled in the air after elevated temperature.

**Figure 7 polymers-15-02127-f007:**
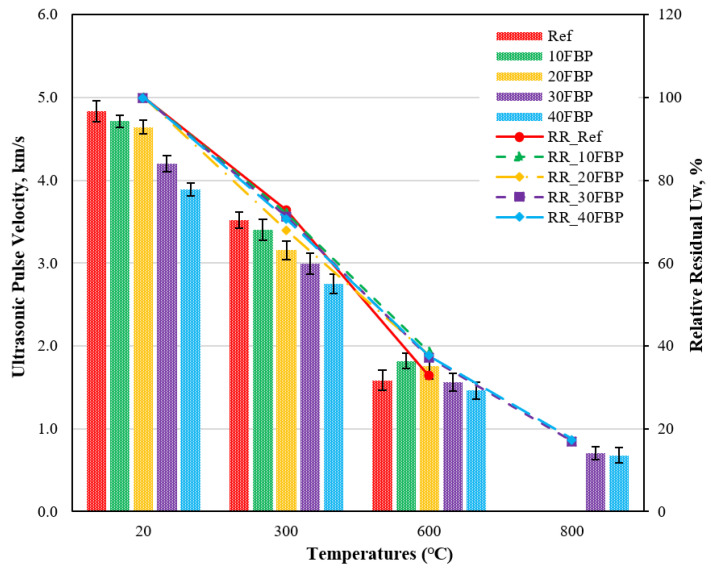
Ultrasonic pulse velocity results of geopolymer samples cooled in the water after elevated temperature.

**Figure 8 polymers-15-02127-f008:**
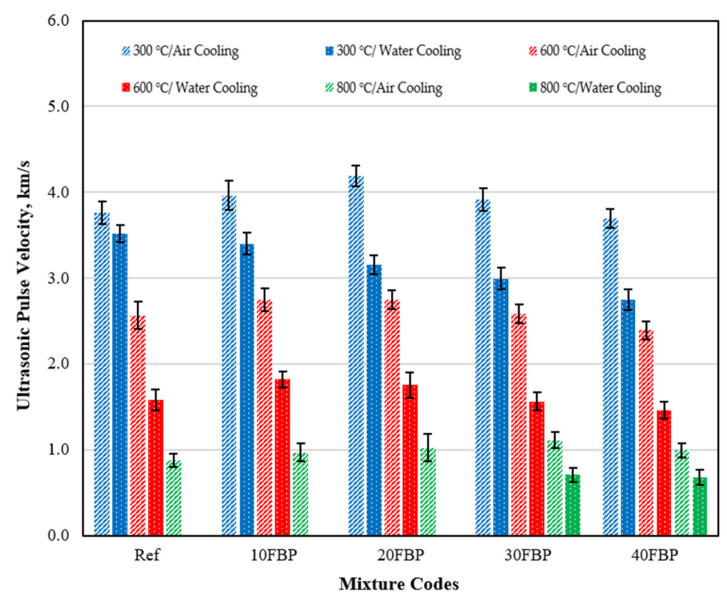
Ultrasonic pulse velocity results of the geopolymer samples cooled in air and water after elevated temperature.

**Figure 9 polymers-15-02127-f009:**
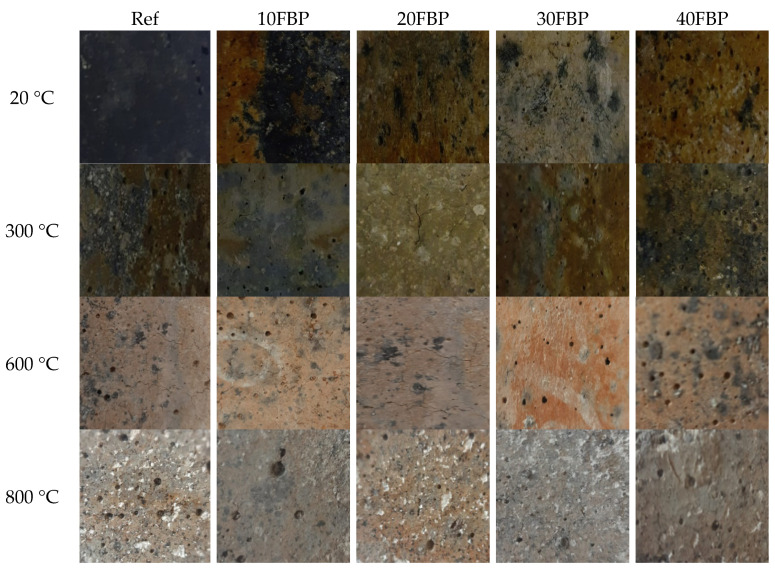
The surface appearances of the geopolymer composite samples after air-cooling.

**Figure 10 polymers-15-02127-f010:**
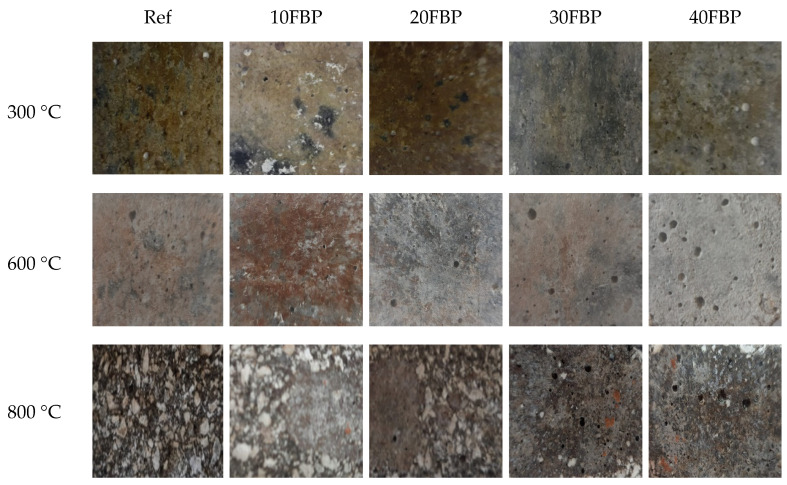
The surface appearances of the geopolymer composite samples after the water-cooling.

**Figure 11 polymers-15-02127-f011:**
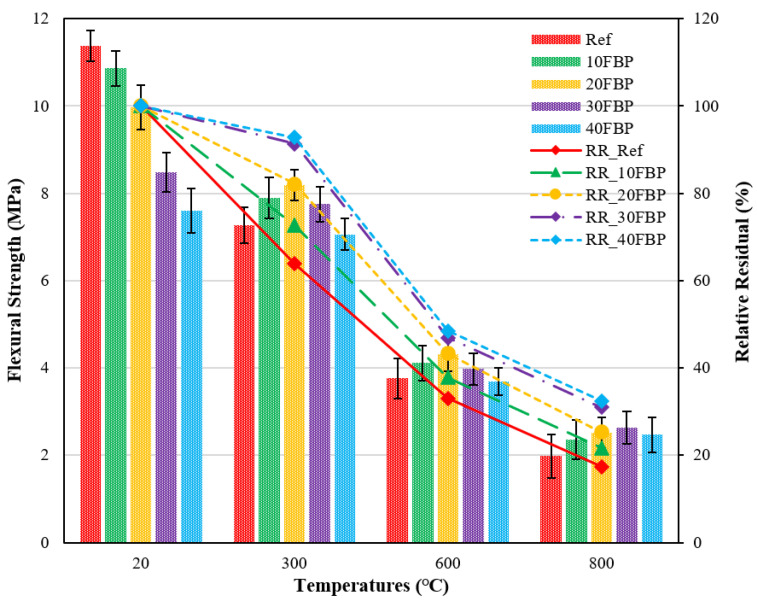
Flexural strength results of geopolymer composite samples cooled in the air after elevated temperature.

**Figure 12 polymers-15-02127-f012:**
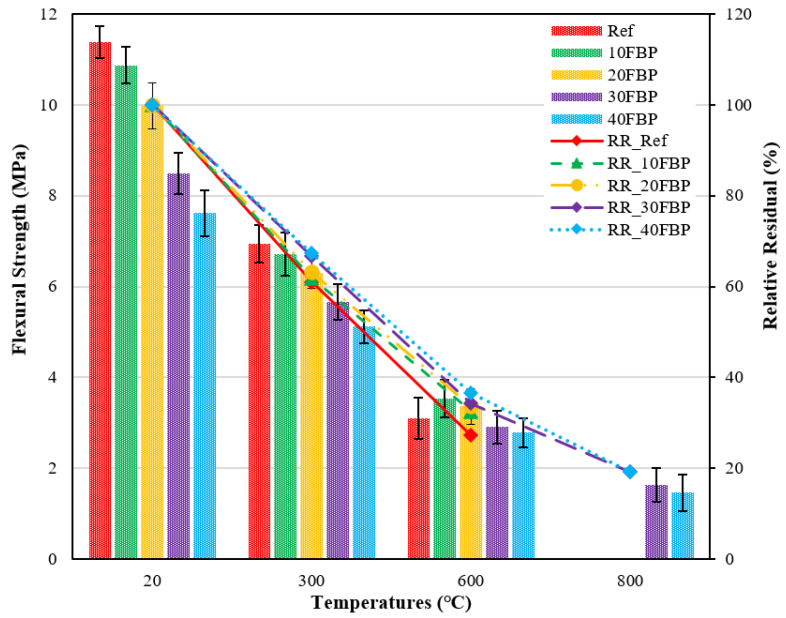
Flexural strength results of geopolymer composite samples cooled in water after elevated temperature.

**Figure 13 polymers-15-02127-f013:**
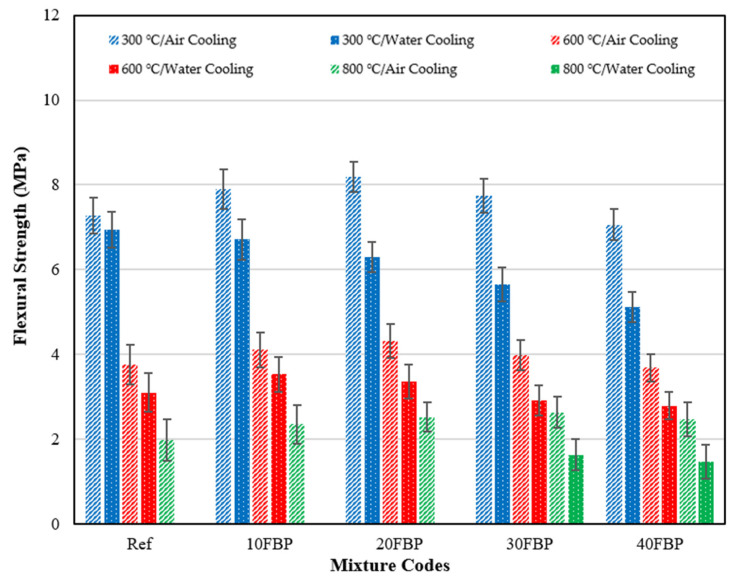
Flexural strength results of the geopolymer composite samples cooled in air and water after elevated temperature.

**Figure 14 polymers-15-02127-f014:**
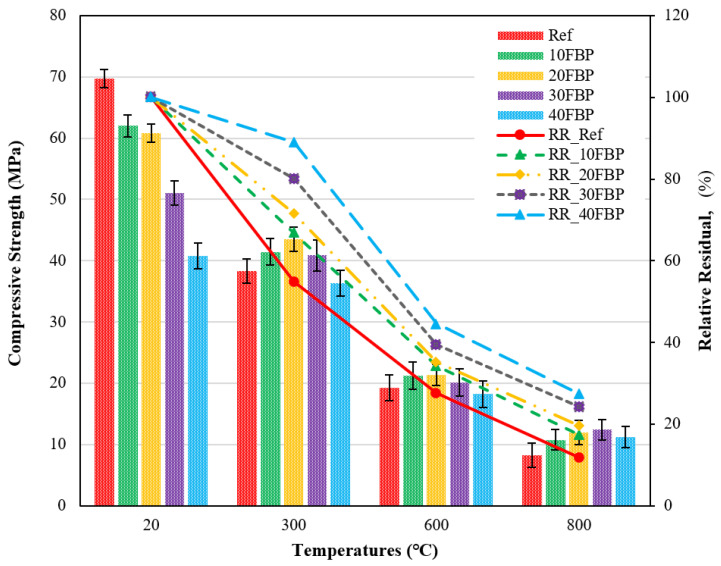
Results of geopolymers cooled in the air after elevated temperature.

**Figure 15 polymers-15-02127-f015:**
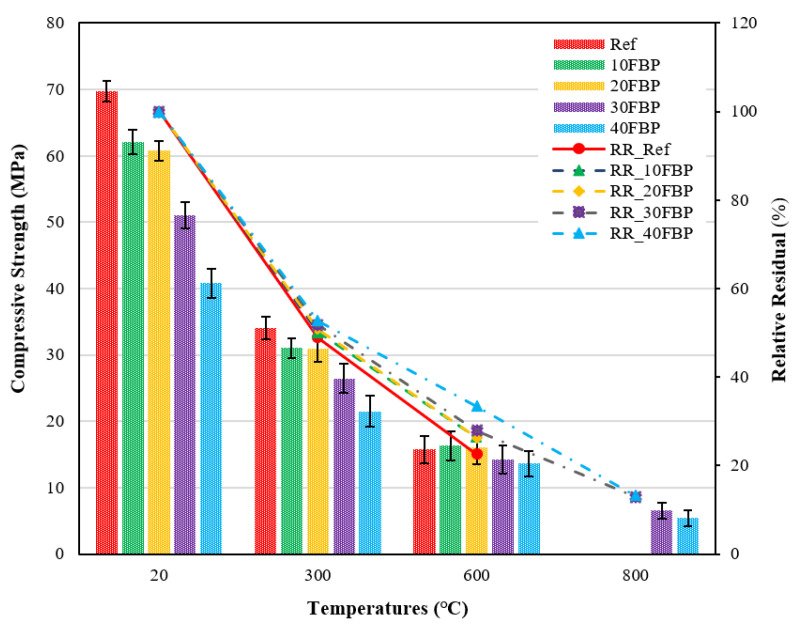
Results of geopolymers cooled in water after elevated temperature.

**Figure 16 polymers-15-02127-f016:**
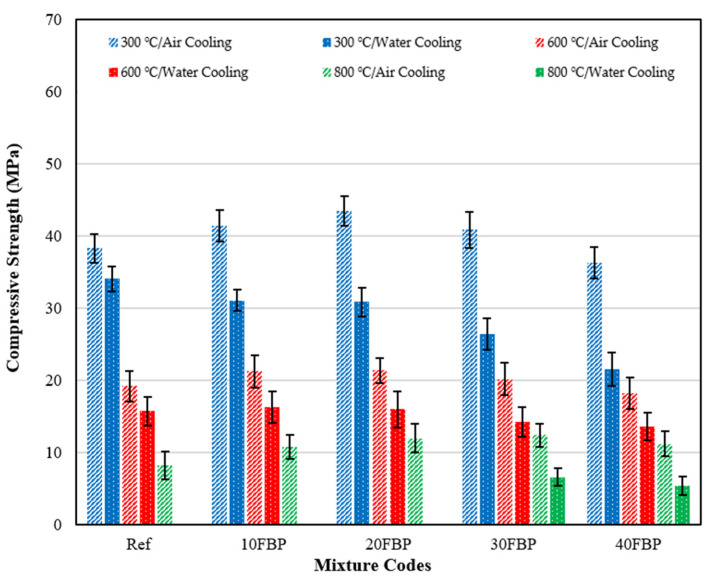
The compressive strength results of the geopolymer composite samples cooled in air and water after elevated temperature.

**Figure 17 polymers-15-02127-f017:**
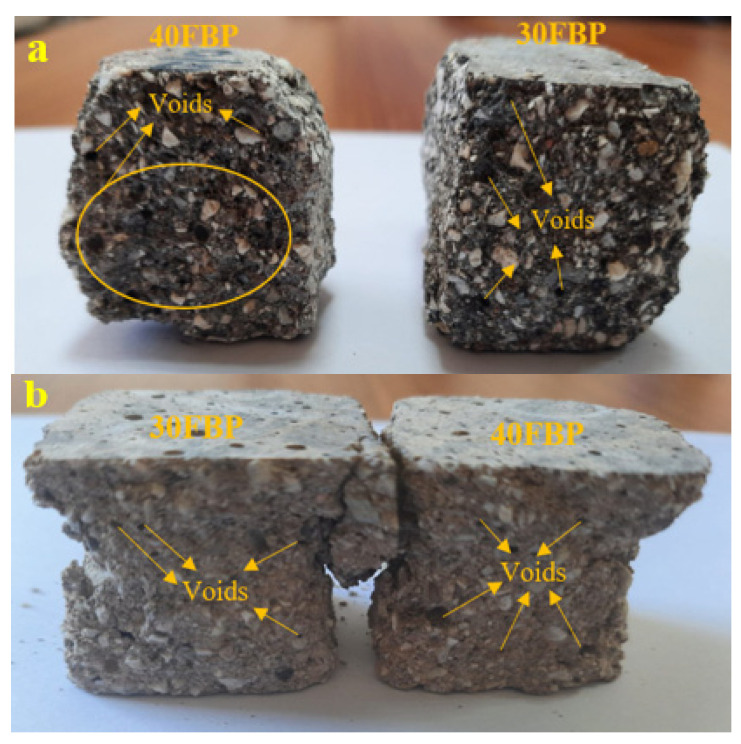
The images of 30FBP and 40FBP samples after 800 °C; (**a**) cooling in water and (**b**) cooling in air.

**Figure 18 polymers-15-02127-f018:**
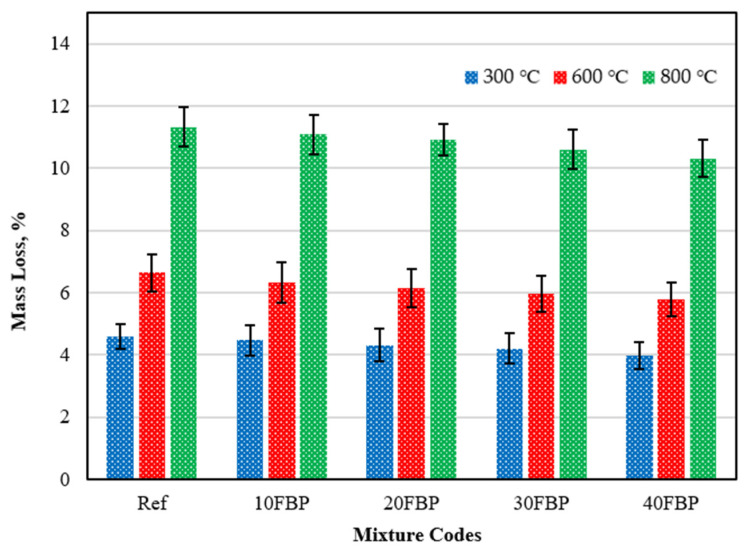
The mass loss results of the geopolymer composite samples cooled in the air after exposure to elevated temperature.

**Figure 19 polymers-15-02127-f019:**
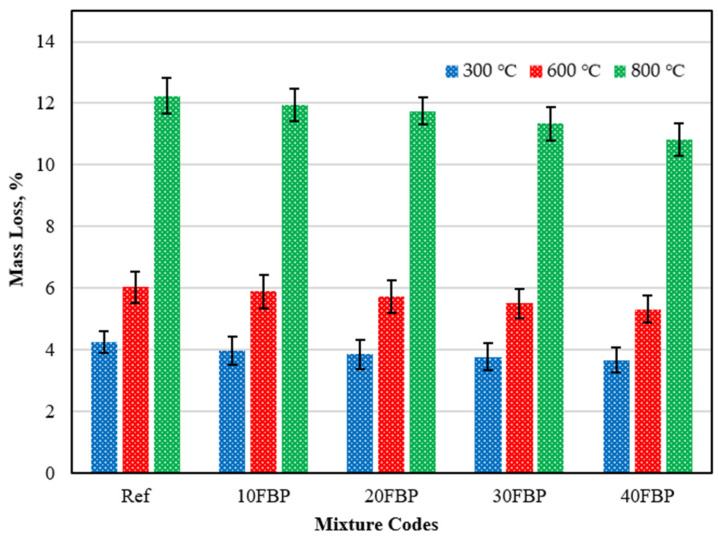
The results of the geopolymers cooled in water after exposure to elevated temperature.

**Figure 20 polymers-15-02127-f020:**
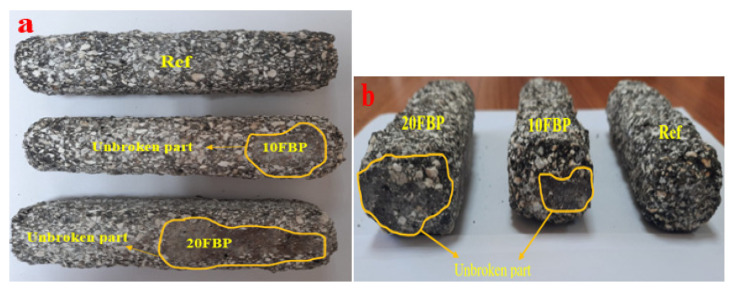
The images of the Ref, 10FBP and 20FBP samples after 800 °C; (**a**) the top, (**b**) side views.

**Figure 21 polymers-15-02127-f021:**
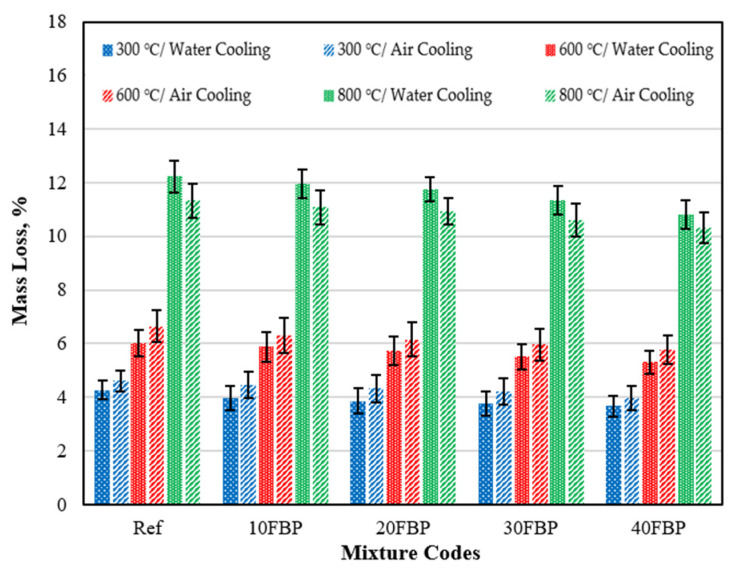
The mass loss results of the geopolymer composite samples cooled in air and water after elevated temperature.

**Figure 22 polymers-15-02127-f022:**
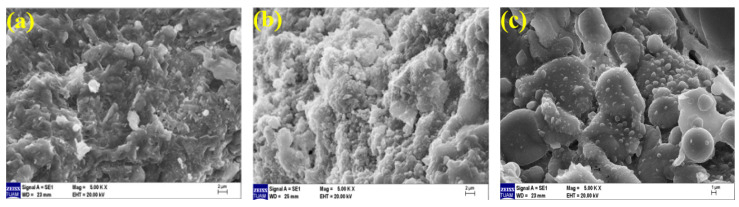
SEM images of Ref samples cooled in the air after elevated temperatures of (**a**) 20 °C, (**b**) 300 °C, and (**c**) 600 °C.

**Figure 23 polymers-15-02127-f023:**
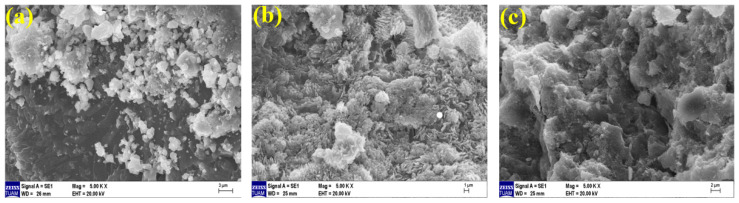
SEM images of 20FBP samples cooled in the air after elevated temperatures of (**a**) 20 °C, (**b**) 300 °C, and (**c**) 600 °C.

**Figure 24 polymers-15-02127-f024:**
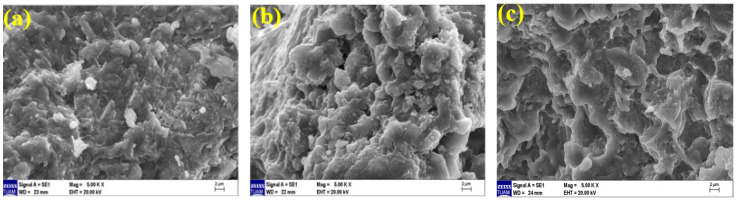
SEM images of Ref samples cooled in water after elevated temperatures of (**a**) 20 °C, (**b**) 300 °C, and (**c**) 600 °C.

**Figure 25 polymers-15-02127-f025:**
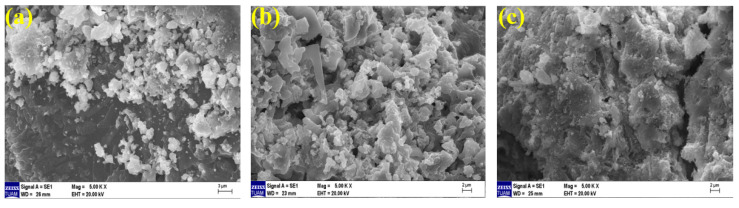
SEM images of 20FBP samples cooled in water after elevated temperatures of (**a**) 20 °C, (**b**) 300 °C, and (**c**) 600 °C.

**Table 1 polymers-15-02127-t001:** The physical and chemical components of BFS and FBP.

Chemical Components (%)	BFS	FBP
SiO_2_	36.11	51.45
Al_2_O	15.19	37.45
Fe_2_O_3_	0.63	2.13
CaO	36.09	0.70
MgO	5.63	0.62
Na_2_O	0.30	0.31
K_2_O	0.82	0.36
SO_3_	1.22	0.29
TiO_2_		4.82
Physical properties		
Specific gravity (unitless)	2.89	2.53
Blaine fineness (cm^2^/g)	5222	2615
Loss on ignition (%)	1.08	6.86

**Table 2 polymers-15-02127-t002:** The preliminary trial results with 100% slag-based-geopolymer composite mortars.

Serial Names	Molarity	Curing Time	Curing Temperature (°C)	Compressive Strength (MPa)
1	12	24	60	18.07
2	14	24	60	25.12
3	16	24	60	27.23
4	12	48	60	34.08
5	14	48	60	34.87
6	16	48	60	37.84
7	12	24	80	38.19
8	14	24	80	52.83
9	16	24	80	49.49
10	12	48	80	44.13
11	14	48	80	45.72
12	16	48	80	58.00
13	12	24	100	46.62
14	14	24	100	60.00
**15**	**16**	**24**	**100**	**69.72**
16	12	48	100	50.07
17	14	48	100	49.32
18	16	48	100	68.08
19	12	24	110	55.47
20	14	24	110	67.72
21	16	24	110	69.67
22	12	48	110	53.45
23	14	48	110	55.77
24	16	48	110	62.36

**Table 3 polymers-15-02127-t003:** The mixture proportions of the different geopolymer composite mortars.

Mixture Code	BFS (g)	FBP (g)	Water (g)	Sand (g)	NaOH (g)	Molarity (M)
Ref	450	0	192.60	1350	144	16
10FBP	405	45	192.60	1350	144	16
20FBP	360	90	192.60	1350	144	16
30FBP	315	135	192.60	1350	144	16
40FBP	270	180	192.60	1350	144	16

**Table 4 polymers-15-02127-t004:** The quality of concrete on the bases of the U_pv_ speed.

U_pv_ (km/s)	The Quality of Concrete
>4.5	Excellent
3.5–4.5	Good
3.0–3.5	Doubtful
2.0–3.0	Poor
<2.0	Very Poor

## Data Availability

Not applicable.

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
