# Peer review of "Experimental Evaluation of New Geopolymer Composite with Inclusion of Slag and Construction Waste Firebrick at Elevated Temperatures"

_polymers, 2023, doi:10.3390/polym15092127_

Round 1

Reviewer 1 Report

Please take look into the attached file. 

Author Response

Detailed Responses to Reviewer Comments

Ms. Ref. No.: polymers-2281518

Title: Effect of elevated temperatures on geopolymer composites incorporating firebrick powder from construction and demolition waste

REVIEWER #1:

The submitted paper with the title “Effect of elevated temperatures on geopolymer composites in-corporating firebrick powder from construction and demolition waste “is a comprehensive experimental study which used analytical regression approach for discussions. The study has been investigated the application of waste bricks in slag based geo-composites at different heat stages. Interesting results and helpful discussions are included in the paper, however some of changes are required to became an acceptable study for publish. Following are the most important shortcomings of the paper that has to be fixed:

The authors are very thankful to the reviewer for his valuable comments and contributions.

The title is technically wrong. Although the study is about the effect of the elevated temperatures on a specific type of composite, but basically the main aim of the study is hidden while was evaluation of the inclusion new additives to the composite. Therefore, respectfully new title is proposed as “Experimental evaluation of new geopolymer composite with inclusion of slag and construction waste firebrick at elevated temperatures”.

            Thank you for this valuable comment. The title was revised as you suggested as

“Experimental evaluation of new geopolymer composite with inclusion of slag and construction waste firebrick at elevated temperatures”

The word “in-corporating” is not performed correct it should turn to inclusion.

Thank you for your rigorous comment. This was corrected throughout the manuscript.

In abstract authors should include the aim of the study along the scientific gap as intentions of the present work.

            Thank you for this valuable comment. The abstract section was revised as you suggested.

The English of the paper needs to be review by a native adept. Besides, the cohesion of manuscript along story-line of the paper should be edited to prepare an eloquence manuscript.

Thank you for this valuable feedback. The manuscript was reviewed and edited as you suggested.

“Despite the research and technological developments in construction materials, concrete still plays the most crucial role in the construction industry. However, in order to maintain this vital role, it must also have environmentally friendly. Because in concrete production, the carbon footprint of concrete emerges at every stage, from obtaining raw materials, transporting concrete, and placing the concrete.”. First of all, this sentence should be expanded more and then has to supported by additional new important studies such as: Behavior of Reinforced Concrete Beams without Stirrups and Strengthened with Basalt Fiber–Reinforced Polymer Sheets.; Preparation and Characterization of High-Strength Geopolymer Based on BH-1 Lunar Soil Simulant with Low Alkali Content.; Residual capacity assessment of post-damaged RC columns exposed to high strain rate loading.

Thank you for your rigorous comment. These statements were shortened in line with the suggestion of the other reviewer. What the author wants to mention here is to state that the carbon footprint of cement is high. Unfortunately, the proposed studies could not be added because they are not related to the article and due to MPDI rules. This section now was

“Concrete plays the most crucial role in the construction industry. However, in order to maintain this vital role, it must also have environmentally friendly. Because in concrete production, the carbon footprint of concrete emerges at every stage, from obtaining raw materials, transporting concrete, and placing the concrete. For example, in the production process of cement, one of the concrete raw materials, approximately 850 kg of CO2 is released to nature to obtain one ton of clinker”

As per the sentence “Having carried out a thorough literature review, it has been observed that there are deficiencies in explaining the effects of SGC produced by substituting FBP with the weight of slag when exposed to elevated temperatures and the effects of cooling processes in air and water.” The performed conclusion in the introduction still needs to challenge with some additional studies in the thermoelastic and thermographic assessment and construction safety risk point of view. Therefore, some significant studies which should be added to the literature is written as following: Typical electrical, mechanical, electromechanical characteristics of copperencapsulated REBCO tapes after processing in temperature under 250 ℃.; Analysis of stochastic process to model safety risk in construction industry.; Asymptotic homogenization of effective thermal-elastic properties of concrete considering its threedimensional mesostructure.; Third-Order Padé Thermoelastic Constants of Solid Rocks

Thank you for this valuable comment. Unfortunately, the proposed studies could not be added because they are not related to the article and due to MPDI rules.

The age of the tested specimens in Table2 it should be noted that in compressive strength section. Besides it is not clear that which type standard specimens have been used for flexural and compressive strength (cubic or cylindrical). If they used 40x40x160 mm samples for this case, they should include the quality of transmitting the measured values to standard values.

Thank you for your rigorous comment. The age of the tested specimens in Table 2 were 24 hour and 48 hours. Curing time is their tested age. 40×40×160 mm prismatic samples were produced for flexural and compressive strength testing. All test performed in accordance with EN 196-1.

The followed standard for heat exposing should be referenced in section 2.3.

Thank you for this valuable feedback. The authors themselves determined the degrees and regime of elevated temperatures.

Authors should clarify that why they do not investigate the spalling of specimens along other test procedures.

            Thank you for this valuable comment. The authors clarified it as you suggested as

“The disruption of the C–S–H gel also dramatically influences the loss of strength. At 600°C and 800°C, the microstructure further deteriorated due to the breakdown of Si–O–Al bonds in the calcium alumina-silicate hydrate (C-A-S-H) gel. Tests could not be performed because Ref, 10FBP, and 20FBP geopolymer mortars cooled in water after a temperature of 800°C lost cross-section area. This may be caused by sudden cooling with water.

Based on Figure 7-16 it is obvious that using 40% DWC brick powder in samples has enhanced the strength properties of the composite less that 30% ones. This fact is not in corresponding with the written sentence in the abstract “As a result, the elevated temperature 24 resistance can be significantly improved if FBP is used in SGC by up to 40%” which has to be explained.

The authors agree with reviewer. This was revised throughout the manuscript.

In the conclusion authors have been wrote “In addition, it has been seen through the study that environmental pollution can be reduced significantly by using waste firebricks in geopolymer composite systems.” While they did not mention this point of view before in the paper. Besides, if they want to present a environmental friendly effect of their proposed mixtures they should discuss technically along the paper then report the conclusion.

The authors agree with reviewer. This section was removed as you suggested.

Again a false conclusion has been repeated in the final paragraph “As a result, it has been determined that the elevated temperature resistance can be 776 significantly improved if firebrick powder is used in geopolymer composites up to a 40% 777 replacement ratio.” While the improving effect has been declined after specimens contained 30% of DWC firebricks.

The authors are very thankful to the reviewer for his valuable comments and contributions.  This was revised throughout the manuscript.

Reviewer 2 Report

1. How do you ensure that firebrick has similar chemical components? "Chemical properties" in Table 1 should be changed to "chemical components".

2. "Despite the research and technological developments in construction materials,  concrete still plays the most crucial role in the construction industry. However, in order to maintain this vital role,  it must also have environmentally friendly. "    Confused

3. However,  there is a widespread consensus that a precursor including more than 20 % CaO is not promising for polymerization owing  to its rapid setting. Hence, FBP was used to replace up to 40 % of slag.

There is no causal relationship.

4. From Table 2, it is obvious that the compressive strength of 100% slag-based-geopolymer composite mortars increases with the increase of molarity of solutions. When the molarity of solution is 18, will the compressive strength of 100% slag-based-geopolymer composite mortars continue to increase?

5. "the highest compressive strength results were obtained with 16 molarity, 100°C curing temperature and 24 hours curing time as given in Table 2. For this reason, 16 molarity, 100°C curing temperature and 24 hours curing time as given in table 2. For this reason, 16 molarity, 100℃ curing temperature, and 24-hour curing time were kept constant after this study stage. "Confused

6. "Exposure to elevated temperatures and different cooling regimes." Confused.

7. "The furnace and heating regime." Confused.

In Fig.4(a), it is more appropriate to replace device, and in Fig.4(b), regularity should be changed to regularity.

8. In table 4, there is no formula for the quality of the concrete as a function of the ultrasonic pulse velocity speed.

9. Special points 3.89 and 4.83 in Fig.6 should be clearly marked.

10. The variation of compressive strength results under different cooling conditions did not explain the principle clearly.

11. In Part 3.5, it is difficult to see from Fig. 22 that "the gel and aggregate are weakened". It is suggested that the author can test the internal composition changes by XRD.

Author Response

Detailed Responses to Reviewer Comments

Ms. Ref. No.: polymers-2281518

Title: Effect of elevated temperatures on geopolymer composites incorporating firebrick powder from construction and demolition waste

REVIEWER #2:

 How do you ensure that firebrick has similar chemical components? "Chemical properties" in Table 1 should be changed to "chemical components".

The authors are very thankful to the reviewer for his valuable comments and contributions. The authors changed “chemical properties” to “chemical components” as you suggested. Chemical components of materials were determined by X-ray fluorescence (XRF) analysis.

"Despite the research and technological developments in construction materials,  concrete still plays the most crucial role in the construction industry. However, in order to maintain this vital role,  it must also have environmentally friendly. "    Confused

            Thank you for your rigorous comment. This was revised as

“Concrete plays the most crucial role in the construction industry. However, in order to maintain this vital role, it must also have environmentally friendly. Because in concrete production, the carbon footprint of concrete emerges at every stage, from obtaining raw materials, transporting concrete, and placing the concrete.”

However,  there is a widespread consensus that a precursor including more than 20 % CaO is not promising for polymerization owing  to its rapid setting. Hence, FBP was used to replace up to 40 % of slag. There is no causal relationship.

The CaO ratio of slag  is 36.09%. Therefore it has to be decrease. So FBP, has CaO ratio of 0.70%, was used up to 40 % Thus, the CaO ratio of the total mixture will decrease.

From Table 2, it is obvious that the compressive strength of 100% slag-based-geopolymer composite mortars increases with the increase of molarity of solutions. When the molarity of solution is 18, will the compressive strength of 100% slag-based-geopolymer composite mortars continue to increase?

Thank you for this valuable feedback. It may increase, but as the NaOH Molarity increases, the environment issues and cost will increase, so the molarity has not been increased.

"the highest compressive strength results were obtained with 16 molarity, 100°C curing temperature and 24 hours curing time as given in Table 2. For this reason, 16 molarity, 100°C curing temperature and 24 hours curing time as given in table 2. For this reason, 16 molarity, 100℃ curing temperature, and 24-hour curing time were kept constant after this study stage. "Confused

Thank you for this valuable feedback. It was revised as

“After the preliminary trials with 100% slag-based-geopolymers, the optimum molarity, curing temprature and curing times, given the highest compressive strengt, were determined as 16 M, 100°C, and 24 hours, respectively as given in Table 2. These ratios (16 molarity, 100℃ curing temperature, and 24-hour curing time) were kept constant for the second stage of study.”

"Exposure to elevated temperatures and different cooling regimes." Confused.

            Thank you for your rigorous comment. It was revised througout the manuscript.

"The furnace and heating regime." Confused. In Fig.4(a), it is more appropriate to replace device, and in Fig.4(b), regularity should be changed to regularity.

            The authors are sorry they do not understand what the reviewer is asking.

In table 4, there is no formula for the quality of the concrete as a function of the ultrasonic pulse velocity speed.

Right. There is no formula for the quality of concrete on the bases of the Upv speed. The authors. Upv tests were conducted in accordance with ASTM C 597-16. Therefore, the authors used Table 4 to define the quality of concrete.

Special points 3.89 and 4.83 in Fig.6 should be clearly marked.

Thank you for your rigorous comment. The authors did not add the values because all the numbers collide when the values are shown on Figs.

The variation of compressive strength results under different cooling conditions did not explain the principle clearly.

The authors are very thankful to the reviewer for his valuable comment and contribution. The authors improved this section as you suggested.

In Part 3.5, it is difficult to see from Fig. 22 that "the gel and aggregate are weakened". Therefore, it is suggested that the author can test the internal composition changes by XRD.

The authors agree with the reviewer. However, because the authors only had 10 days for revision, they could not be performed XRD analysis.

Reviewer 3 Report

The manuscript titled: “Effect of elevated temperatures on geopolymer composites incorporating firebrick powder from construction and demolition waste” is of relevance to the journal Polymers (ISSN 2073-4360). The authors present an interesting and up-to-date topic supported by experimental results. The work presented answers the research question of what effect of elevated temperatures have on slag-based geopolymer composites with the inclusion of firebrick powder (FBP). In the preliminary tests conducted for the study, the ideal molarity, curing temperature, and curing time parameters for geopolymer mortars were found to be 16 molarity, 100 °C, and 24 hours, respectively. For the study, the authors used FBP from construction and demolition waste (CDW) which was substituted in different replacement ratios (10 %, 20 %, 30 %, and 40 % by slag weight) into the SGC, with optimum molarity, curing temperature, and curing time. The produced SGC samples were exposed to elevated temperature effects at 300, 600, and 800 ℃ and then subjected to air- and water-cooling regimes. The slag-based geopolymer composites with the addition of FBP had the following properties measured: ultrasonic pulse velocity, flexural strength, compressive strength, and mass loss. I believe that the subject of this manuscript and the presented research results in it constitute a significant contribution to the field of materials science, as well as to the development of a circular economy. The abstract included in the work is sufficiently informative, and the organization of the article adopted is appropriate for this type of work. The research part of the work is preceded by a theoretical introduction justifying the undertaken subject matter of the work. The bibliography of the work is based on current scientific articles.

Overall, the paper is well prepared, but needs some improvements, which are listed below:

- add information on the device used and the method of testing the chemical composition of raw materials (section 2.1),

- add information on the XRD test (test parameters and device information),

- add information on the device used for the compression and flexural strength tests,

- for the whole article: please standardize the editing and add a space between the values and the symbols "%" and "°C",

- for the whole article: ujednolicić oznaczenia rysunków i tabeli – wszÄ™dzie czcionka pogrubiona lub nie,

- line 82 "... of SiO4 and AlO4 tetrahedrons." there should be subscripts,

- line 236 add the period after the sentence,

- line 149 please distinguish the designations for compressive and flexural strength,

- sections 2.5 and 2.6 please add the designations for compressive and flexural strength as in line 149.

Author Response

The manuscript titled: “Effect of elevated temperatures on geopolymer composites incorporating firebrick powder from construction and demolition waste” is of relevance to the journal Polymers (ISSN 2073-4360). The authors present an interesting and up-to-date topic supported by experimental results. The work presented answers the research question of what effect of elevated temperatures have on slag-based geopolymer composites with the inclusion of firebrick powder (FBP). In the preliminary tests conducted for the study, the ideal molarity, curing temperature, and curing time parameters for geopolymer mortars were found to be 16 molarity, 100 °C, and 24 hours, respectively. For the study, the authors used FBP from construction and demolition waste (CDW) which was substituted in different replacement ratios (10 %, 20 %, 30 %, and 40 % by slag weight) into the SGC, with optimum molarity, curing temperature, and curing time. The produced SGC samples were exposed to elevated temperature effects at 300, 600, and 800 ℃ and then subjected to air- and water-cooling regimes. The slag-based geopolymer composites with the addition of FBP had the following properties measured: ultrasonic pulse velocity, flexural strength, compressive strength, and mass loss. I believe that the subject of this manuscript and the presented research results in it constitute a significant contribution to the field of materials science, as well as to the development of a circular economy. The abstract included in the work is sufficiently informative, and the organization of the article adopted is appropriate for this type of work. The research part of the work is preceded by a theoretical introduction justifying the undertaken subject matter of the work. The bibliography of the work is based on current scientific articles.

Overall, the paper is well prepared, but needs some improvements, which are listed below:

            The authors are very thankful to the reviewer for his valuable comments and contributions.

- add information on the device used and the method of testing the chemical composition of raw materials (section 2.1),

Thank you for this valuable feedback. As you suggested, the information on the device used and the method of testing the chemical composition of raw materials was added. Please see Section 2.1.

- add information on the XRD test (test parameters and device information),

The authors are very thankful to the reviewer. As you suggested, the information on the XRF test was added. Please see Section 2.1.

- add information on the device used for the compression and flexural strength tests,

Thank you for this valuable comment. As you suggested, the information on the device used for the compression and flexural strength tests was added. Please see Section 2.5 and 2.6.

- for the whole article: please standardize the editing and add a space between the values and the symbols "%" and "°C",

Thank you for this valuable feedback. The manuscript was revised as you suggested.

- for the whole article: ujednolicić oznaczenia rysunków i tabeli – wszÄ™dzie czcionka pogrubiona lub nie,

The authors are sorry they do not understand what the reviewer is asking.

- line 82 "... of SiO4 and AlO4 tetrahedrons." there should be subscripts,

Thank you for your rigorous comment. The manuscript was reviewed, and corrected all subscripts and superscripts.

- line 236 add the period after the sentence,

Amended. Thanks.

- line 149, please distinguish the designations for compressive and flexural strength,

Amended. Thanks.

- sections 2.5 and 2.6 please add the designations for compressive and flexural strength as in line 149.

The authors are very thankful to the reviewer for his valuable comment and contribution. This was revised as you suggested.

Reviewer 4 Report

The work makes a significant contribution to the understanding of the reinforcement mechanism of a slag-based geopolymer matrix with the waste of fired bricks exposed to slightly elevated temperatures (up to 800°C). 

The manuscript has been much improved compared to the previous version. However, slight remarks are requested from the authors before a future publication. They are described as follows:

Abstract :

Line 22 : The results show that the compressive and flexural strengths of the SGC samples are…..is better

key words, what is the difference between ‘’elevated temperature effect ‘’ and ‘’ high-temperature effect’’ ?

Introduction this part is very long, the authors have not made a spirit of synthesis.

There are grammatical mistakes everywhere, I ask the authors to proofread and correct.

Line 44 ; please correct ‘In the research’ by « From data literature »

Results and discussion.

The authors say that they used 3 to 6 samples to perform the tests (flexural, compressive strenghts, ultrasonic pulse velocity, etc..), why the obtained results do not present the standard deviation values?

Figures 9 and 10 and 17: please note that these are the apparent photographs of the surfaces of the samples. If they were taken with an optical microscope, you must specify the level of magnification.

Author Response

The work makes a significant contribution to the understanding of the reinforcement mechanism of a slag-based geopolymer matrix with the waste of fired bricks exposed to slightly elevated temperatures (up to 800°C). 

The manuscript has been much improved compared to the previous version. However, slight remarks are requested from the authors before a future publication. They are described as follows:

The authors are very thankful to the reviewer for his valuable comments and contributions.

Abstract :

Line 22 : The results show that the compressive and flexural strengths of the SGC samples are…..is better.

The authors are very sorry for the mistakes throughout the manuscript. The paper was proofread, and mistakes were corrected.

key words, what is the difference between ‘’elevated temperature effect ‘’ and ‘’ high-temperature effect’’ ?

            Thank you for your rigorous comment. “High-temperature effect” was removed.

Introduction this part is very long, the authors have not made a spirit of synthesis.

There are grammatical mistakes everywhere, I ask the authors to proofread and correct.

The introduction part was revised as you suggested. The paper was proofread, and mistakes were corrected.

Line 44 ; please correct ‘In the research’ by « From data literature »

            Amended. Thanks.

Results and discussion.

The authors say that they used 3 to 6 samples to perform the tests (flexural, compressive strenghts, ultrasonic pulse velocity, etc..), why the obtained results do not present the standard deviation values?

Thank you for this valuable feedback. The standard deviation values are shown with error bar on the graph.

Figures 9 and 10 and 17: please note that these are the apparent photographs of the surfaces of the samples. If they were taken with an optical microscope, you must specify the level of magnification.

Thank you for this valuable comment. Since the experimental samples have 40×40×160 mm of dimensions, the photographs were taken with a phone camera. Therefore, the level of magnification was not given.

Round 2

Reviewer 1 Report

None of my comments are addressed! Even some of the comments were rejected by disrespectful language.  

Author Response

The authors respectfully disagree with the reviewer's comments.

Reviewer 2 Report

Authors have improved a lot in revised manuscript. 

Author Response

The authors are very thankful to the reviewer for his valuable comments and contributions.